# Tetrahedron Splatting for 3D Generation

**Chun Gu**[1]  **Zeyu Yang**[1]  **Zijie Pan**[1]  **Xiatian Zhu**[2]  **Li Zhang**[1]*
[1]School of Data Science, Fudan University    [2]University of Surrey

https://fudan-zvg.github.io/tet-splatting

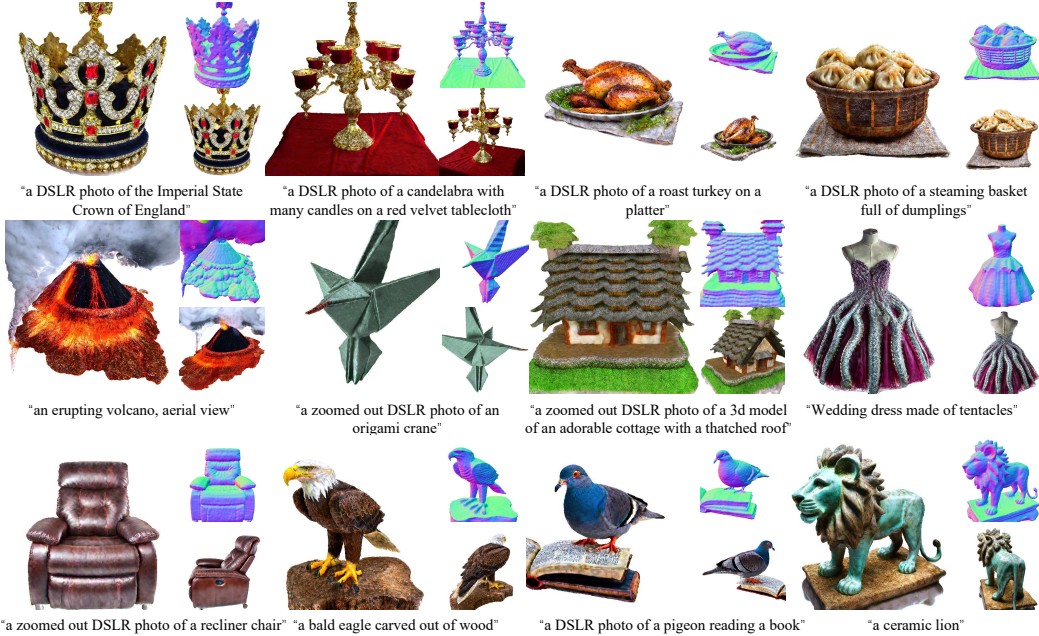

"a DSLR photo of the Imperial State Crown of England"

"a DSLR photo of a candelabra with many candles on a red velvet tablecloth"

"a DSLR photo of a roast turkey on a platter"

"a DSLR photo of a steaming basket full of dumplings"

"an erupting volcano, aerial view"

"a zoomed out DSLR photo of an origami crane"

"a zoomed out DSLR photo of a 3d model of an adorable cottage with a thatched roof"

"Wedding dress made of tentacles"

"a zoomed out DSLR photo of a recliner chair"  "a bald eagle carved out of wood"

"a DSLR photo of a pigeon reading a book"

"a ceramic lion"

Figure 1: 3D assets generated by our proposed *TeT-Splatting*.

## Abstract

3D representation is essential to the significant advance of 3D generation with 2D diffusion priors. As a flexible representation, NeRF has been first adopted for 3D representation. With density-based volumetric rendering, it however suffers both intensive computational overhead and inaccurate mesh extraction. Using a signed distance field and Marching Tetrahedra, DMTet allows for precise mesh extraction and real-time rendering but is limited in handling large topological changes in meshes, leading to optimization challenges. Alternatively, 3D Gaussian Splatting (3DGS) is favored in both training and rendering efficiency while falling short in mesh extraction. In this work, we introduce a novel 3D representation, Tetrahedron Splatting (*TeT-Splatting*), that supports easy convergence during optimization, precise mesh extraction, and real-time rendering *simultaneously*. This is achieved by integrating surface-based volumetric rendering within a structured tetrahedral grid while preserving the desired ability of precise mesh extraction, and a tile-based differentiable tetrahedron rasterizer. Furthermore, we incorporate eikonal and normal consistency regularization terms for the signed distance field to improve

*Li Zhang (lizhangfd@fudan.edu.cn) is the corresponding author.

38th Conference on Neural Information Processing Systems (NeurIPS 2024).

generation quality and stability. Critically, our representation can be trained without mesh extraction, making the optimization process easier to converge. Our *TeT-Splatting* can be readily integrated in existing 3D generation pipelines, along with polygonal mesh for texture optimization. Extensive experiments show that our *TeT-Splatting* strikes a superior tradeoff among convergence speed, render efficiency, and mesh quality as compared to previous alternatives under varying 3D generation settings.

# 1 Introduction

Table 1: Comparison of different representations for 3D generation.

| Representation | NeRF [28] | 3DGS [13] | DMTet [40] | TeT-Splatting (**Ours**) |
|---|---|---|---|---|
| **Precise mesh extraction** | | | ✓ | ✓ |
| **Easy convergence** | ✓ | ✓ | | ✓ |
| **Real-time rendering** | | ✓ | ✓ | ✓ |
| **Representative method** | *DreamFusion* [32], *Magic3D* [18] | *DreamGaussion* [46], *GSGEN* [5] | *Fantasia3D* [3], *RichDreamer* [34] | *Ours* |

Automatic 3D content generation is revolutionizing fields such as virtual reality, augmented reality, video games, and industrial design. This technology can significantly enhance user experiences and streamline creative processes for reducing time demands and simplifying the complexities associated with creating high-quality 3D assets.

3D representations (*e.g.*, Neural Radiance Field [28] (NeRF)) play an essential role in recent advancements in 3D generation, along with the Score Distillation Sampling (SDS) technique objective [32] for exploiting off-the-shelf 2D diffusion models [11, 37, 38, 1]. Although serving as a pioneer representation, NeRF is significantly limited due to its intensive computational demands, particularly when paired with high-resolution 2D diffusion models. Moreover, its density-based volumetric rendering struggles with accurate mesh extraction, which is crucial for practical applications.

By utilizing a signed distance field and Marching Tetrahedra for differentiable mesh extraction, DMTet [40] enables efficient high-resolution rendering and precise mesh extraction, overcoming the limitations of the NeRF approach. In cases, it becomes a favored choice [18, 33, 3]. However, DMTet is limited in its ability to handle large topological changes in meshes, as it can only backpropagate to the zero-level set of the signed distance field, constraining its geometry convergence during optimization. As a workaround, a two-stage 3D generation pipeline has been adopted that initially utilizes NeRF for rapid geometry convergence and then transitions to DMTet for detailed refinement [18]. However, transitioning from NeRF to DMTet often results in a degradation of quality, as the strengths of each representation are not fully leveraged throughout the entire optimization process.

Alternatively, recent methods have introduced 3D Gaussian Splatting [13] (3DGS) into the optimization process, significantly enhancing efficiency. For example, DreamGaussian [46] utilizes 3DGS but acknowledges that meshes directly generated from 3DGS can be blurry, and the mesh extraction process often results in unsatisfactory surfaces with visible holes [45]. Moreover, the text-to-3D process with 3DGS suffers from instability due to its unstructured nature and the densification process.

In this work, we introduce *TeT-Splatting*, a novel all-round 3D representation that integrates surface-based volumetric rendering into the tetrahedral grid, while preserving precise mesh extraction through Marching Tetrahedra. It supports easy convergence during optimization, precise mesh extraction, and real-time rendering *simultaneously*, enabling high-fidelity 3D generation effectively (Figure 1). Drawing inspiration from 3DGS [13], we design a tile-based fast differentiable rasterizer for real-time rendering, efficiently handling the alpha-blending of projected 2D splats from 3D tetrahedra. These splats are blended based on opacity values derived from the signed distance field within each tetrahedron as in NeuS [49]. To further increase efficiency, we include a pre-filtering process to remove nearly transparent tetrahedra, reducing the number of tetrahedra necessary for splatting. Moreover, we introduce eikonal and normal consistency regularization terms to refine the signed distance field, which helps stabilize the optimization process and prevents the common issue of debris in the optimization with DMTet. In Table 1 we compare the features of different 3D representations.

Our **contributions** are fourfold: **(i)** Introducing a novel 3D representation, *TeT-Splatting*, that integrates synergistically volumetric rendering into a tetrahedral grid; **(ii)** Designing a fast differentiable rasterizer for tetrahedra; **(iii)** Forming a generic two-stage 3D generation pipeline that initially leverages *TeT-Splatting* for geometry optimization, and then transitions it to polygonal mesh for texturing; **(iv)** Extensive evaluations demonstrating the superior tradeoff of our method among easy convergence, real-time rendering, and precise mesh extraction over alternative representations (InstantNGP [29], DMTet [40], and 3DGS [13]) under a variety of settings with different diffusion priors.

## 2 Related work

**3D representation** Since the introduction of Neural Radiance Field (NeRF) [28], NeRF has become a foundational technique in the field of 3D reconstruction. It employs volumetric rendering to enable 3D optimization with only 2D supervision. Despite its significance, NeRF faces major issues, such as slow rendering speeds and high memory usage. To address these problems, several research[43, 39, 29, 2, 13] have developed novel variants of the radiance field, focusing on faster training and rendering and using less computing resources. Diverging from the path of NeRF, DMTet [40] has introduced an approach based on Marching Tetrahedra and surface rendering by differentiable rasterization [14], offering much faster rendering speed. Recently, 3D Gaussian Splatting [13] (3DGS) has unified NeRF-like alpha-blending with tile-based rasterization, achieving high performance in both quality and rendering speed. In this paper, our proposed *TeT-Splatting* takes inspiration from the structured tetrahedral grid in DMTet [40] and incorporates tile-based rasterization from 3DGS [13], utilizing tetrahedra for splatting. *TeT-Splatting* achieves high converge and rendering speed while preserving precise mesh extraction through Marching Tetrahedra.

**3D generation** The data-driven 2D diffusion models [11, 37, 38, 1] have demonstrated unprecedented success in image generation. However, the transition to direct 3D generation [31, 12, 10, 24, 8, 58, 6, 52, 19, 15, 45] faces formidable challenges, as this research line often fails to generate high-quality 3D assets limited by the lack of training data. To circumvent these issues, some works [20, 41, 23, 42, 21, 25, 50] train 2D diffusion models to make them have 3D awareness. However, discrete and sparse 2D images still cannot offer sufficient 3D information. In this context, DreamFusion [32] first introduced score distillation sampling (SDS) loss to leverage 2D diffusion priors for 3D generation. Subsequent studies [47, 57, 59, 54, 51, 17, 26, 44, 48] have aimed to improve the SDS loss, enhancing both the fidelity and stability of 3D generation. Moreover, several efforts [55, 5, 56, 4, 16, 42, 34, 22] have been made to improve the quality and multi-view consistency of 3D models by integrating a wider array of diffusion priors. Despite these advancements, some methods are hindered by the significant computational demands due to the usage of NeRF [28], which limits the effective use of high-resolution diffusion priors. Additionally, other mesh-based models [3, 16, 34] encounter issues with instability and slow convergence due to the nature of surface rendering. By contrast, our *TeT-Splatting* facilitates the use of high-resolution diffusion priors and ensures efficient updates, thanks to its volumetric rendering and tile-based differentiable rasterizer.

## 3 TeT-Splatting

### 3.1 Deformable tetrahedral grid

In this section, we will start with an introduction to the deformable tetrahedral grid, which is the geometric primitive for the proposed representation. The deformable tetrahedral grid is first employed in DefTet [7] and then extended in DMTet [40] to approximate the implicit surface by assigning each vertex an SDF value. Specifically, this structure considers a tetrahedral mesh composed of $N$ vertices and $K$ tetrahedra, denoted as $(V_T, T)$, where $V_T = \{\boldsymbol{v}_n | n \in 1, \dots, N\}$ signifies the positions of vertices, and $T = \{t_k | k \in 1, \dots, K\}$, with each $t_k$ representing the indices $(a_k, b_k, c_k, d_k)$ of four vertices $(\boldsymbol{v}_{a_k}, \boldsymbol{v}_{b_k}, \boldsymbol{v}_{c_k}, \boldsymbol{v}_{d_k})$ that form a tetrahedron. Utilizing the SDF value associated with each vertex $\boldsymbol{v}_n$, denoted by $f_n$, a signed distance field is established by interpolating the SDF values within each tetrahedron. DMTet [40] has developed a method for mesh extraction from the tetrahedral grid by assigning one or two triangles to each tetrahedron that intersects the zero-level set of the signed distance field, known as Marching Tetrahedra (MT). Employing a differentiable triangular rasterizer, it attains a remarkable rendering speed while maintaining minimal memory consumption.

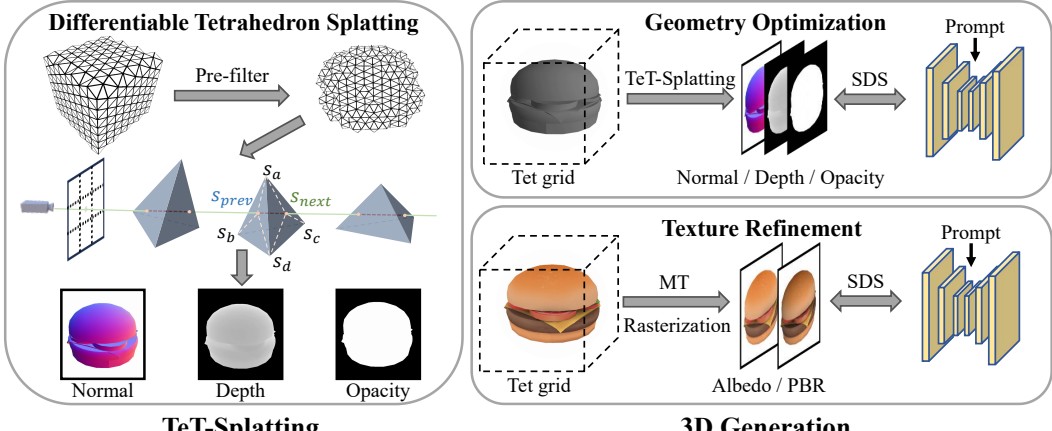

Figure 2: **Left: An overview of *TeT-Splatting*.** To produce the final renderings, we first pre-filter and remove nearly transparent tetrahedra, then project the remaining ones into 2D splats. These are blended based on opacity values derived from the SDF values at specific pixel intersections. **Right: *TeT-Splatting* for 3D generation.** We employ *TeT-Splatting* in the initial stage of the 3D generation pipeline and subsequently transition it to polygonal mesh for texture optimization.

However, a particular limitation of the MT is that only the parameters associated with tetrahedra intersecting the zero-level set of the signed distance field can be updated during optimization. This restriction poses challenges in managing large topological changes and often causes the optimization to stuck in the undesired shape in the early stage. In contrast, NeRF is less affected by such instability thanks to its volumetric nature. Many prior works [18, 33] have employed NeRF in 3D generation. These works typically adopt a two-stage pipeline that starts with the volumetric representation to swiftly achieve a coarse model with low-resolution diffusion priors and then transitions to a polygonal mesh for further refinement with high-resolution diffusion priors. However, these approaches are often hindered by the slow optimization and inaccurate geometry brought by volume rendering. The inaccurate geometry would lead to obvious degradation after mesh extraction.

## 3.2 Differentiable tetrahedron splatting

In this work, we present a unified representation that combines the precise mesh extraction via the tetrahedral grid and the efficient optimization of volumetric rendering. Inspired by 3D Gaussian Splatting [13] (3DGS), we also integrate the tile-based rasterizer into our framework to facilitate real-time rendering. 3DGS enhances rendering efficiency through rasterization and ensures efficient optimization via alpha-blending by projecting 3D Gaussians to 2D splats followed by fast alpha-blending. However, 3DGS relies on unstructured 3D Gaussians as rendering primitives, necessitating carefully designed densification processes and learning rates to manage the highly noisy SDS loss. In contrast, the tetrahedral grid is structured while its vertices can only deform in a local region and are connected with neighbors to form tetrahedra. We explore treating tetrahedron as rendering primitive of the splatting process to perform alpha-blending. Moreover, we can directly extract polygonal mesh through Marching Tetrahedra from the tetrahedral grid, while the mesh extracted [46] from 3DGS may result in an unsatisfactory surface with visible holes.

Next, we will elaborate on how we realize differentiable tetrahedron splatting through alpha-blending. Consider a pixel on the image plane, along with its corresponding ray in 3D space. To perform alpha-blending, we need first determine the intersected tetrahedra between the ray and the tetrahedral grid. For a single tetrahedron $t$ with vertices $(\boldsymbol{v}_a, \boldsymbol{v}_b, \boldsymbol{v}_c, \boldsymbol{v}_d)$ and SDF values $(f_a, f_b, f_c, f_d)$, we can project the vertices onto the image plane, resulting in four overlapped triangles that form a 2D tetrahedron splat. Intersection with the tetrahedron $t$ is equal to the intersection with the four triangles. The position and SDF value of an intersection point can be calculated using barycentric coordinates (see Appendix A for details). Different from 3DGS [13], we consider the opacity of tetrahedra instead of Gaussians. Note that a ray can only have two intersection points with a tetrahedron, we denote their SDF values as $f_{\text{prev}}$ and $f_{\text{next}}$ in depth order. Then the opacity of the tetrahedron $t$ can be derived

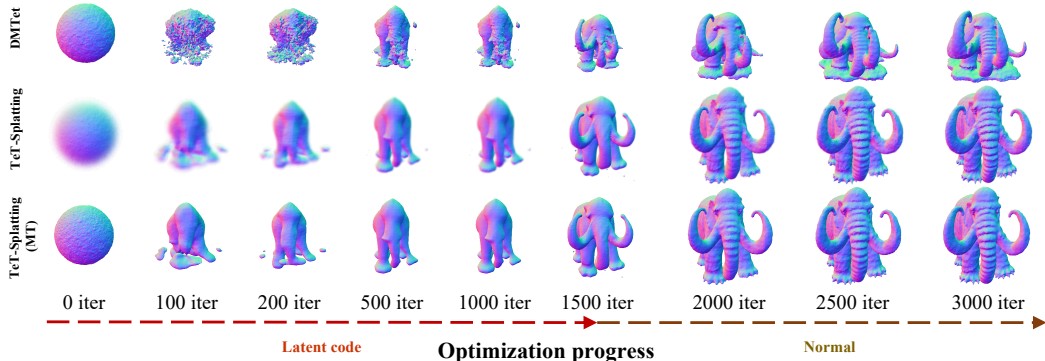

Figure 3: **Normal map comparison during optimization of 3D generation.** We utilize DMTet and *TeT-Splatting* as 3D representations in the geometry modeling stage of the RichDreamer [34]. The first two rows show normal maps obtained from DMTet and *TeT-Splatting* during optimization. *TeT-Splatting* achieves more stable and smooth optimization, while DMTet becomes fragmented initially and gets stuck in an undesirable shape. The third row shows the normal maps of meshes extracted from the signed distance field of *TeT-Splatting* via Marching Tetrahedra [40] (MT). As optimization progresses, *TeT-Splatting*'s behavior aligns with rendering through MT.

in NeuS [49] manner:

$$\alpha = \max\left(\frac{\Phi_s(f_{\text{prev}}) - \Phi_s(f_{\text{next}})}{\Phi_s(f_{\text{prev}})}, 0\right), \quad (1)$$

where $\Phi_s(x) = (1 + e^{-sx})^{-1}$, and the $s$ value controls the steepness of the conversion. Following Voxurf [53], we update $s$ manually for each iteration $i$: $s = i/s_{\text{ratio}} + s_{\text{start}}$. The final normal map $\mathcal{N}$, depth map $\mathcal{D}$ and opacity map $\mathcal{O}$ are derived by alpha-blending $N$ sequentially ordered tetrahedra from front to back:

$$\{\mathcal{N}, \mathcal{D}, \mathcal{O}\} = \sum_{i \in N} T_i \alpha_i \{\boldsymbol{n}_i, \boldsymbol{z}_i, 1\}, \quad T_i = \prod_{j=1}^{i-1}(1 - \alpha_j), \quad (2)$$

where $\boldsymbol{n}$ denotes the per-tetrahedron normal and $\boldsymbol{z}_i$ denotes the average depths of four vertices.

**Pre-filtering** To conserve computational resources, the tetrahedra with low opacity will be filtered. Depending on different intersection points, the opacity of a tetrahedron can take different values. We can establish the upper bound of the opacity, denoted as $\alpha_{max}$, by replacing $s_{prev}$ and $s_{next}$ in Eq. 1 with the maximum and minimum SDF values of four vertices. Tetrahedra with $\alpha_{max}$ less than a predefined threshold $T_f = \frac{1}{255}$ are filtered to ensure that only tetrahedra with significant enough contribution to the alpha-blending are included in the subsequent splatting process.

**Per-tetrahedron normal** As discussed in Section 3.1, the tetrahedral grid establishes a signed distance field by interpolating the SDF values within each tetrahedron. This interpolation is a linear combination of the SDF values of four vertices. Correspondingly, the barycentric coordinates of an arbitrary point with respect to the four vertices of the tetrahedron exhibit a linear correlation with its spatial position. This ensures that the gradient $\boldsymbol{g}$ of the SDF within the tetrahedron results in a constant vector (see Appendix A for details). The normal vector $\boldsymbol{n}$ of the tetrahedron is thus obtained by normalizing this gradient.

**Relationship between DMTet and TeT-Splatting** During optimization, DMTet employs Marching Tetrahedra to extract polygonal mesh from the tetrahedral grid and subsequently renders through triangular rasterization [14]. Consequently, only a limited number of tetrahedra are involved in each single rendering process. In contrast, *TeT-Splatting* employs volumetric rendering, which allows all visible tetrahedra within the view frustum that have sufficient weight in alpha-blending to contribute to the final renderings. Moreover, the rendering process in *TeT-Splatting* is fully differentiable, enabling a single optimization step to influence a significantly larger number of parameters compared to DMTet. Figure 3 presents a comparative analysis of convergence speeds between DMTet and *TeT-Splatting* within the same Text-to-3D pipeline. As observed, *TeT-Splatting* achieves rapid convergence, whereas DMTet exhibits slower topological changes and gets stuck in an undesirable shape. Furthermore, as the inverse standard deviation $s$ in Eq. 1 increases, the curve of $\Phi_s(x)$ becomes steeper, causing

$\alpha$ to approach 1 under conditions that $s_{prev} > 0$ and $s_{next} < 0$ and to approach 0 otherwise. This behavior (Figure 3) aligns with the rendering process of DMTet [40] through Marching Tetrahedra, where only the tetrahedra intersecting the zero-level set of the signed distance field are visible.

## 3.3 Fast differentiable rasterizer for tetrahedra

We implement a tile-based differentiable rasterizer for tetrahedra with custom CUDA kernel building upon the framework of 3DGS [13]. Similarly, we begin by dividing the screen into tiles and culling tetrahedra that do not overlap with the view frustum. We then replicate the tetrahedra based on the number of tiles they overlap and sort them by their tile ID and the average depth of each tetrahedron's vertices, using a fast GPU radix sort [27]. Note that the per-tile sorting in 3DGS is not equivalent to per-pixel ordering. Differently, we maintain a short resorting window [36] of size $N_w$ for each pixel re-sort the primitives based on the results of per-tile sorting using the insertion sort. Due to the structured nature of the tetrahedral grid, we find that the sorting error is almost eliminated with a window size of 5 under a grid resolution of 256. The operations after re-sorting for alpha-blending are the same as in 3DGS, except for the computation of $\alpha$, which we have already given in Eq. 1.

# 4 3D generation with TeT-Splatting

In this section, we introduce our 3D generation pipeline and discuss various settings, aiming at validating the effectiveness of *TeT-Splatting* in 3D generation. As shown in Figure 2, our pipeline is divided into two stages: first get a detailed geometry with *TeT-Splatting* and then transition to polygonal mesh through Marching Tetrahedra [40] for texture optimization. We begin by describing our overall 3D model (Section 4.1), and then detail the regularizations (Section 4.2) and diffusion priors (Section 4.3) used in our experiments.

## 4.1 3D modeling

**Geometry stage** We employ a hash grid [29] $\Phi_g$ with parameter $\Theta_g$ to encode the signed distance field and deformation which allows each vertex in a tetrahedral grid to deform in a certain range. $\Phi_g$ is initialized to a spherical shape. Given a randomly sampled camera, our tetrahedron rasterizer produces renderings of the normal map, depth map, and opacity map.

**Texture stage** Given the well-optimized signed distance field from the geometry stage, we convert it into a polygonal mesh through Marching Tetrahedra. To texture the polygonal mesh, we employ the physically based rendering (PBR) pipeline proposed by Nvdiffrec [30]. Please refer to [30, 9, 34] for details. We use another hash grid $\Phi_t$ with parameter $\Theta_t$ to encode the spatially varying materials of the surface: albedo, roughness, metallic, and bump. Finally, given a specific environment lighting and a randomly sampled camera, we can obtain the renderings of the albedo map and PBR map.

## 4.2 Regularization

**Eikonal loss** To ensure a proper signed distance field, we employ an eikonal term that regularizes the SDF gradient $g$ in each tetrahedron: $\mathcal{L}_{\text{eik}} = \sum_k \left( \|g_k\|_2 - 1 \right)^2$.

**Normal consistency loss** Inspired by the normal consistency loss for triangle meshes, we adapt this approach to tetrahedra. While we have designed a per-tetrahedron normal, we project these tetrahedral normals onto vertices and enhance the consistency of the signed distance field by regularizing the cosine similarity between normals of adjacent vertices connected by edges: $\mathcal{L}_{\text{nc}} = \sum_i \left( 1 - \cos \left( n_{e_{i1}}, n_{e_{i2}} \right) \right)$, where $e_{i1}$ and $e_{i2}$ represent the vertices forming edge $e_i$.

## 4.3 Diffusion priors

To validate the capability of *TeT-Splatting* for 3D generation, we employ two types of diffusion priors: the vanilla RGB-based diffusion priors and the rich diffusion priors proposed in RichDreamer [34].

**Vanilla RGB-based diffusion priors** Vanilla RGB-based diffusion models represent diffusion models that can generate RGB images from a given prompt. For both geometry and texture stages, we utilize SDS loss to leverage 2D diffusion priors from Stable Diffusion [37]:

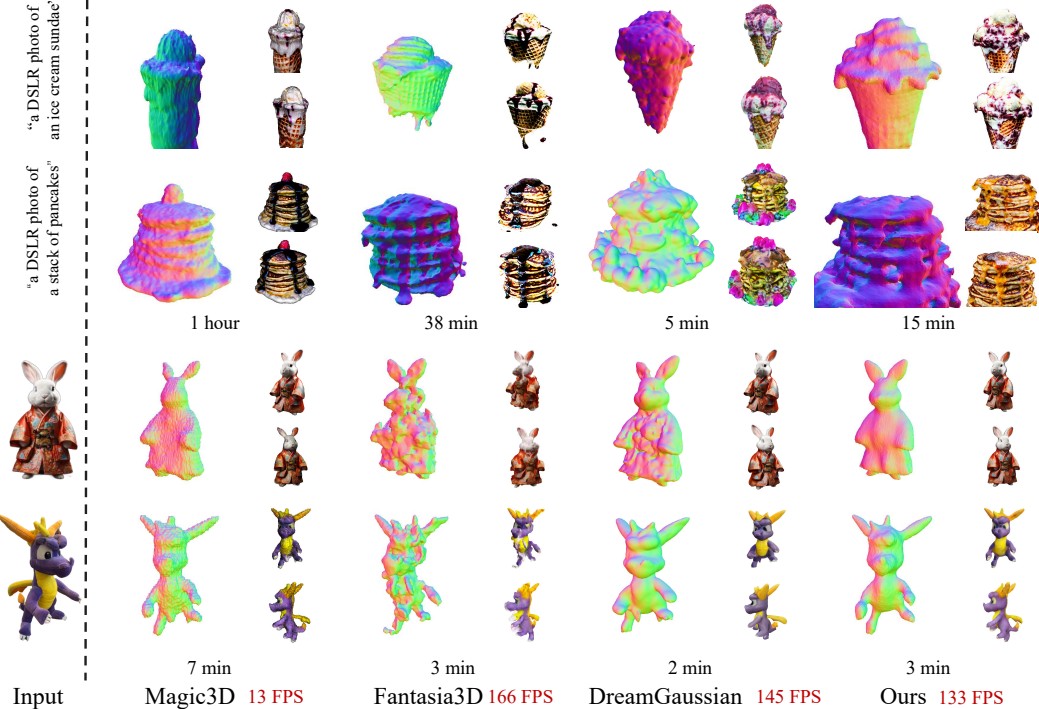

Figure 4: Qualitative comparison on 3D generation using vanilla RGB-based diffusion priors. We present visual comparisons of the rendered RGB maps and color maps from various 3D generation methods. The methods, arranged from left to right, are: Magic3D, Fantasia3D, DreamGaussian, and Ours. The comparison is conducted across two tasks, text-to-3D and image-to-3D, with results shown from top to bottom, respectively. Additionally, for each method, we provide the training time and the rendering speed (FPS) for the first stage of the process.

$\nabla_{\Theta} \mathcal{L}_{\text{SDS}} = \mathbb{E} \left[ \omega(t)(\epsilon_{\phi}(\mathcal{I}; y, t) - \epsilon) \frac{\partial z}{\partial \mathcal{I}} \frac{\partial \mathcal{I}}{\partial \Theta} \right]$, where $\omega(t)$ is a weighting function, $z$ denotes the VAE latent code, and $\epsilon_{\phi}(\mathcal{N}; y, t)$ represents the noise estimated by the UNet $\epsilon_{\phi}$.

**Rich diffusion priors**  The sole use of vanilla diffusion priors often leads to issues such as multi-face Janus problem, domain gap between image diffusion model and normal map while using normal maps as input of diffusion models, and inaccuracies in material decomposition. To this end, we utilize the rich diffusion priors proposed in RichDreamer [34] to handle high-fidelity 3D generation. Specifically, for geometry optimization, we combine a vanilla Stable Diffusion with a Normal-Depth diffusion model, which generates multi-view normal and depth maps from a given text prompt, represented as: $\mathcal{L}_{\text{SDS}} = \mathcal{L}_{\text{SDS-Normal}}^{\text{SD}} + \mathcal{L}_{\text{SDS-ND}}^{\text{ND}}$. For texture optimization, we combine a vanilla SD with a Depth-conditioned Albedo diffusion model, capable of producing multi-view albedo maps from a given text prompt, represented as: $\mathcal{L}_{\text{SDS}} = \mathcal{L}_{\text{SDS-RGB}}^{\text{SD}} + \mathcal{L}_{\text{SDS-Albedo}}^{\text{Albedo}}$.

In summary, the final loss function for the geometry stage is defined as: $\mathcal{L}_{\text{geo}} = \mathcal{L}_{\text{SDS}} + \lambda_{\text{eik}} \mathcal{L}_{\text{eik}} + \lambda_{\text{nc}} \mathcal{L}_{\text{nc}}$. For the texture stage, the loss function simplifies to: $\mathcal{L}_{\text{tex}} = \mathcal{L}_{\text{SDS}}$.

## 5 Experiment

In this section, we assess the efficacy of *TeT-Splatting* across two distinct tasks: 3D generation employing vanilla RGB-based diffusion priors and text-to-3D with rich diffusion priors. In Section 5.1, a qualitative evaluation of 3D generation for both text-to-3D and image-to-3D modalities is conducted to demonstrate the superiority of *TeT-Splatting* relative to other representations. To substantiate *TeT-Splatting*'s proficiency in handling high-fidelity generations, we conduct experiments with advanced rich diffusion priors in Section 5.2. Section 5.3 encompasses a series of ablation studies aimed at validating the representation and pipeline. The details of implementation and experimental setting can be found in the Appendix B.

## 5.1 Results with vanilla RGB-based diffusion priors

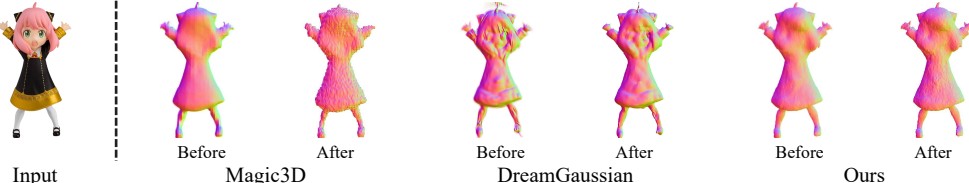

Figure 5: Visualization of normal maps before and after mesh exportation. Note that the normal maps of DreamGaussian [46] are derived from its depth maps.

Focusing on illustrating the effectiveness of the proposed representation, we primarily compare to three competitors: Magic3D [18], Fantasia3D [3] and DreamGaussian [46]. All these competitors employ a two-stage optimization pipeline and leverage Stable Diffusion with SDS loss, but utilize three different representative 3D representations in their initial stages: Instant-NGP [29] (a fast version of NeRF), DMTet [40], and 3DGS [13], respectively. We adapt the diffusion priors of all methods to Stable Zero-1-to-3 [20] for a fair comparison in the image-to-3D task, adding an identical MSE alignment loss. The qualitative evaluations, as shown in Figure 4, illustrate our method's ability to generate more detailed and compact meshes in a relatively short time. In Figure 4, we also report the rendering speed (FPS) of the first stage, at a rendering resolution of 512x512. Although *TeT-Splatting* operates at a lower FPS compared to DMTet (Fantasia3D) and 3DGS (DreamGaussian), it still achieves real-time rendering. Importantly, this lower FPS does not adversely affect the overall generation process, as the primary bottleneck in generation speed lies with the diffusion model.

We also conduct a comparison of the mesh extraction with Magic3D and DreamGaussian, visualized in Figure 5. It reveals that the meshes extracted from Magic3D often do not faithfully replicate the geometries from the first stage due to the imprecise threshold for converting densities to SDF values, and the extracted mesh in DreamGaussian can result in unsatisfactory surfaces with visible holes. In contrast, our method maintains high quality with negligible degradation after mesh extraction.

## 5.2 Results with rich diffusion priors

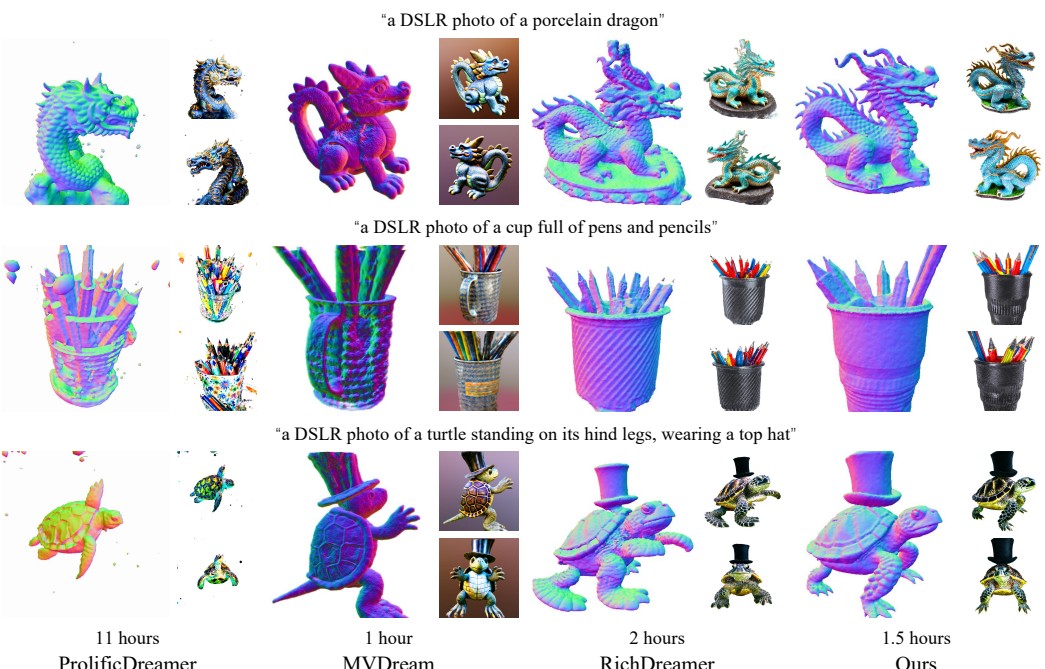

Figure 6: Qualitative comparison on Text-to-3D with rich diffusion priors. We also report the total training time of each method.

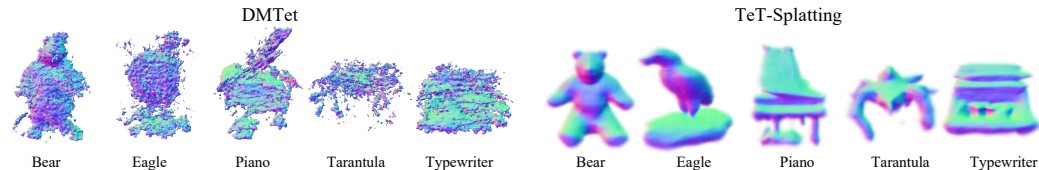

Figure 7: Normal map comparison between DMTet [40] and *TeT-Splatting* in the early training iterations.

"a DSLR photo of a barbecue grill cookingsausages and burger patties"

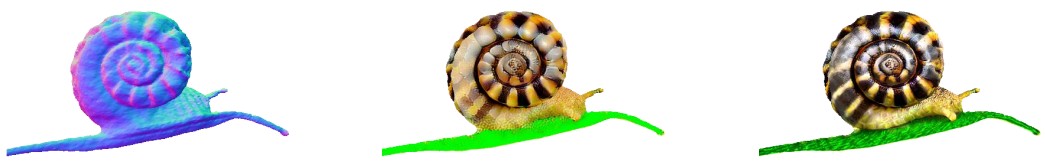

"a DSLR photo of a barbecue grill cookingsausages and burger patties"

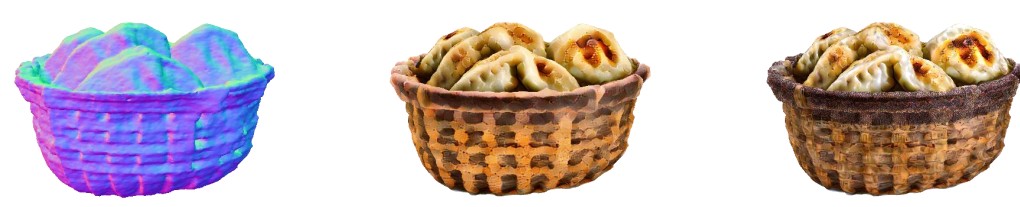

Figure 8: Visualization of the rendered normal, albedo, and PBR map from the generated 3D assets in the second stage.

Our approach is also compatible with state-of-the-art diffusion priors. In this part, we evaluate *TeT-Splatting* on the text-to-3D task, equipped with rich diffusion priors from RichDreamer [34]. A key distinction between our approach and RichDreamer is the use of *TeT-Splatting* as the 3D representation during the geometry stage. We evaluate our method against two SOTA competitors, ProlificDreamer [51] and RichDreamer [34]. As illustrated in Figure 6, our method is capable of handling high-fidelity 3D generation, achieving superior geometric quality with considerably reduced generation times compared to these competitors.

Additionally, we present visualizations of the normal maps from early training iterations in Figure 7. The results from RichDreamer are fragmented at the early iterations due to the use of DMTet, which may harm subsequent optimization and slow convergence. In contrast, *TeT-Splatting* demonstrates rapid and smooth convergence. We employ the same quantitative evaluation method as RichDreamer to assess the quality of geometry and texture.

In Table 2, we report the Geometry CLIP [35] score and Appearance CLIP score. Notably, Rich-Dreamer's prompt list, comprising 113 objects used for scoring, is not publicly available. Consequently, we calculate our scores using an alternative set of prompts (see Appendix B for details). Our method outperforms RichDreamer in terms of CLIP scores and significantly reduces the time required for geometry optimization (40 min vs 70 min).

In Figure 8, we present the decomposed albedo maps of generated 3D assets. Guided by the Depth-conditioned Albedo diffusion model, we achieve natural albedo maps.

### 5.3 Ablation

Under the settings of rich diffusion priors [34], we conduct ablation studies to evaluate our method.

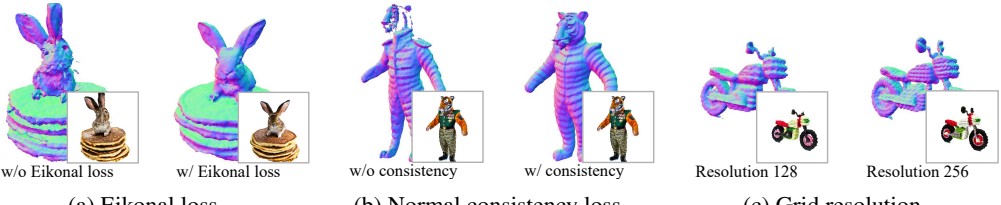

| w/o Eikonal loss | w/ Eikonal loss | w/o consistency | w/ consistency | Resolution 128 | Resolution 256 |
| (a) Eikonal loss | | (b) Normal consistency loss | | (c) Grid resolution | |

Figure 9: **Ablation studies** on eikonal loss, normal consistency loss and the resolution of the tetrahedral grid.

Table 2: **CLIP score comparison**. Results marked with "*" are taken from RichDreamer [34]. Since RichDreamer [34] did not release their prompt list (113 objects), we use our own prompt list (183 objects) for evaluation. See Appendix B for more details.

| | Prolificdreamer [51] | MVDream [42] | RichDreamer [34] | RichDreamer [34] | Ours |
|---|---|---|---|---|---|
| Geometry CLIP score ↑ | 23.3818* | 24.8003* | 25.8820* | 23.0143 | **23.1641** |
| Appearance CLIP score ↑ | 31.8022* | 28.7331* | 31.7099* | 29.2198 | **29.4197** |

**Eikonal loss**   We assess the role of eikonal loss in 3D generation by comparing 3D assets generated with and without it, illustrated in Figure 9a. Models created without eikonal loss tend to develop into undesirable shapes. This issue arises because the SDF values rapidly reach extreme levels and get trapped in local minima when eikonal loss is not applied.

**Normal consistency loss**   Additionally, we assess the importance of normal consistency loss. Figure 9b demonstrates that applying normal consistency loss results in more compact models. This loss can act as a smoothing prior that helps prevent the surface of the model from becoming fragmented.

**Tetrahedral grid resolution**   We investigate the effects of tetrahedral grid resolution on model performance by conducting experiments at resolutions of 128 and 256. Higher resolution yields more detailed geometries, as shown in Figure 9c.

## 6   Limitations

*TeT-Splatting* struggles with modeling high-frequency features, such as texture, because it uses tetrahedra as rendering primitives, which limits the final output by the resolution of the tetrahedral grid. Therefore, we transition it to a polygonal mesh for enhanced texture optimization. The rendering speed of our implemented rasterizer, although operating in real-time, is slower than that of 3DGS. Additionally, using only a pre-filter operation might not fully leverage *TeT-Splatting*'s potential in rendering quality and speed. A similar densification process as in 3DGS could improve this, which we leave for future work.

## 7   Conclusion

In this study, we introduce Tetrahedron Splatting (*TeT-Splatting*), a novel all-round 3D representation that integrates volumetric rendering within a structured tetrahedral grid while preserving precise mesh extraction through Marching Tetrahedra. Equipped with newly designed tile-based fast differentiable tetrahedron rasterizer, *TeT-Splatting* achieves real-time rendering. As showcase, we integrate *TeT-Splatting* in common 3D generation pipeline with polygonal mesh for texture optimization. Extensive experiments under varying 3D generation settings demonstrate *TeT-Splatting*'s superiority in producing high-fidelity 3D content compared to other 3D representations.

## Acknowledgments

This work was supported in part by National Natural Science Foundation of China (Grant No. 62106050 and 62376060) and Natural Science Foundation of Shanghai (Grant No. 22ZR1407500).

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

## A    More implementation details of TeT-Splatting

In this section, we provide additional implementation details about the tetrahedron splatting process.

## A.1 Barycentric coordinates

Recall that projecting a single tetrahedron, encompassing four vertices $\boldsymbol{v}_a, \boldsymbol{v}_b, \boldsymbol{v}_c, \boldsymbol{v}_d \in \mathbb{R}^3$ along with their SDF values $f_a, f_b, f_c, f_d \in \mathbb{R}$, onto a 2D plane results in an array of 2D vectors $\boldsymbol{v}'_a, \boldsymbol{v}'_b, \boldsymbol{v}'_c, \boldsymbol{v}'_d \in \mathbb{R}^2$ representing the coordinates of these vertices on the image plane.

Given a pixel $\boldsymbol{p}$ with coordinate $\boldsymbol{v}'_p \in \mathbb{R}^2$, its barycentric coordinates $u', v' \in \mathbb{R}$ with repsect to a triangle $(\boldsymbol{v}'_a, \boldsymbol{v}'_b, \boldsymbol{v}'_c)$ satisfy $\boldsymbol{v}'_p = (1 - u' - v')\boldsymbol{v}'_a + u'\boldsymbol{v}'_b + v'\boldsymbol{v}'_c$. We can apply the following equations to derive the barycentric coordinates $u', v'$:

$$
\begin{bmatrix} \boldsymbol{v}'_a & \boldsymbol{v}'_b & \boldsymbol{v}'_c \\ 1 & 1 & 1 \end{bmatrix} \begin{bmatrix} 1 - u' - v' \\ u' \\ v' \end{bmatrix} = \begin{bmatrix} \boldsymbol{v}'_p \\ 1 \end{bmatrix}, \tag{3}
$$

$$
\Rightarrow \begin{bmatrix} 1 - u' - v' \\ u' \\ v' \end{bmatrix} = \begin{bmatrix} \boldsymbol{v}'_a & \boldsymbol{v}'_b & \boldsymbol{v}'_c \\ 1 & 1 & 1 \end{bmatrix}^{-1} \begin{bmatrix} \boldsymbol{v}'_p \\ 1 \end{bmatrix}. \tag{4}
$$

If $u', v' \in [0, 1]$, the pixel $\boldsymbol{p}$ is considered inside the triangle.

## A.2 Perspective correction

The calculated barycentric coordinates $u', v'$ are based on 2D projections and require adjustment to reflect the original 3D spatial relationships accurately. This adjustment, known as perspective correction, is necessary because 3D depth information is not preserved in the 2D projection. We perform this correction using:

$$
u = \frac{\frac{u'}{z_b}}{\frac{(1-u'-v')}{z_a} + \frac{u'}{z_b} + \frac{v'}{z_c}}, \qquad v = \frac{\frac{v'}{z_c}}{\frac{(1-u'-v')}{z_a} + \frac{u'}{z_b} + \frac{v'}{z_c}}, \tag{5}
$$

where $z_*$ denotes the depth of each vertex. Subsequently, the SDF value and depth of the 3D position corresponding to this pixel are interpolated:

$$
f_p = (1 - u - v)f_a + u f_b + v f_c, \qquad z_p = \frac{1}{\frac{(1-u'-v')}{z_a} + \frac{u'}{z_b} + \frac{v'}{z_c}}. \tag{6}
$$

## A.3 Gradient of the SDF value inside a tetrahedron

Consider an arbitrary 3D point $\boldsymbol{v}_q$ with SDF value $f_q$ inside the tetrahedron $(\boldsymbol{v}_a, \boldsymbol{v}_b, \boldsymbol{v}_c, \boldsymbol{v}_d)$. We establish the SDF value $f_q$ using the 3D barycentric coordinates $u, v, w$: $f_q = (1 - u - v - w)f_a + u f_b + v f_c + w f_d$. The derivation of $u, v, w$ is similar to 2D case in Section A.1, i.e.,

$$
\begin{bmatrix} \boldsymbol{v}_a & \boldsymbol{v}_b & \boldsymbol{v}_c & \boldsymbol{v}_d \\ 1 & 1 & 1 & 1 \end{bmatrix} \begin{bmatrix} 1 - u - v - w \\ u \\ v \\ w \end{bmatrix} = \begin{bmatrix} \boldsymbol{v}_q \\ 1 \end{bmatrix}, \tag{7}
$$

$$
\Rightarrow \begin{bmatrix} 1 - u - v - w \\ u \\ v \\ w \end{bmatrix} = \begin{bmatrix} \boldsymbol{v}_a & \boldsymbol{v}_b & \boldsymbol{v}_c & \boldsymbol{v}_d \\ 1 & 1 & 1 & 1 \end{bmatrix}^{-1} \begin{bmatrix} \boldsymbol{v}_q \\ 1 \end{bmatrix}. \tag{8}
$$

Therefore, the formula of $f_q$ can be expressed by:

$$
f_q = \begin{bmatrix} f_a & f_b & f_c & f_d \end{bmatrix} \begin{bmatrix} 1 - u - v - w \\ u \\ v \\ w \end{bmatrix} \tag{9}
$$

$$= \begin{bmatrix} f_a & f_b & f_c & f_d \end{bmatrix} \begin{bmatrix} \boldsymbol{v}_a & \boldsymbol{v}_b & \boldsymbol{v}_c & \boldsymbol{v}_d \\ 1 & 1 & 1 & 1 \end{bmatrix}^{-1} \begin{bmatrix} \boldsymbol{v}_q \\ 1 \end{bmatrix}. \tag{10}$$

Moreover, the gradient of the SDF value at $\boldsymbol{v}_q$, denoted by $\boldsymbol{g}$, is straightforwardly derived by differentiating with respect to $\boldsymbol{v}_q$. This gradient is constant across the tetrahedron:

$$\begin{bmatrix} \boldsymbol{g} \\ * \end{bmatrix} = \begin{bmatrix} \boldsymbol{v}_a^\top & 1 \\ \boldsymbol{v}_b^\top & 1 \\ \boldsymbol{v}_c^\top & 1 \\ \boldsymbol{v}_d^\top & 1 \end{bmatrix}^{-1} \begin{bmatrix} f_a \\ f_b \\ f_c \\ f_d \end{bmatrix}. \tag{11}$$

This results in a constant vector within any tetrahedron, providing a consistent gradient that aids in precise mesh extractions and surface optimizations.

## B  More implementation details of 3D generation

In this section, we provide additional implementation details of 3D generation. Note that all experiments are conducted on one NVIDIA RTX A6000 GPU.

### B.1  Geometry stage

Unlike 3DGS [13], we separate operations that are repeated across multiple images from different camera viewpoints within a single training iteration and shift them to the beginning of each iteration, including the inference of SDF values and deformations for each vertex from the hash grid, pre-filtering tetrahedra based on their $\alpha_{max}$, and calculating the per-tetrahedron normal. These pre-processed results are then passed to the rasterizer for rendering a batch of images. Moreover, we implement a coarse-to-fine approach in the pre-filtering process: initially, we establish a tighter axis-aligned bounding box from the pre-filtered tetrahedral grid in the first round and then resize the tetrahedral grid based on this bounding box for a second round of pre-filtering, which enhances the precision of the geometry. For the schedule of the $s$ value in Eq. 1, we set $s_{\text{ratio}} = 5$ and $s_{\text{start}} = 20$, which allows the curve of $\Phi_s(x)$ to be sufficiently steep at the final of optimization. Additionally, we set both $\lambda_{\text{eik}}$ and $\lambda_{\text{nc}}$ to 1000.

### B.2  Evaluation with vanilla RGB-based diffusion priors

This part is implemented based on the threestudio codebase [9] using the settings of Fantasia3D [3]. The tetrahedral grid resolution is set to 128, and the batch size is set to 1.

**Text-to-3D**  For the text-to-3D task, we use Stable Diffusion 2.1 base. The geometry is optimized for 3,000 iterations and the texture for another 1,000 iterations. Following Fantasia3D [3], during the initial training iterations, we concatenate the rendered normal and depth maps to serve directly as the latent code for the diffusion models, facilitating rapid convergence to a basic shape. As Magic3D [18] haven't released their code, we use the implementation from threestudio. To ensure a fair comparison with Fantasia3D [3], we also adopt the threestudio implementation.

**Image-to-3D**  In the image-to-3D task, we use Stable Zero-1-to-3 and adapt our pipeline by incorporating $\Phi_t$ in the initial stage to encode the materials at the center of each tetrahedron. Specifically, we start by inferring the materials at the center of each tetrahedron. Next, we compute the per-tetrahedron PBR color $\boldsymbol{c}$ using the rendering equation for each tetrahedron. This per-tetrahedron PBR color $\boldsymbol{c}$ is subsequently passed to the rasterizer to perform the alpha-blending for the final output: $\mathcal{C} = \sum_{i \in N} T_i \alpha_i \boldsymbol{c}_i$. To further refine the output, we introduce an MSE loss that aligns the rendering color map $\mathcal{C}^r$ and opacity map $\mathcal{O}^r$ at reference view with the provided reference image $\tilde{\mathcal{C}}^r$ and mask $\tilde{\mathcal{O}}^r$: $\mathcal{L}_{\text{ref}} = \lambda_{\text{rgb}}||\mathcal{C}^r - \tilde{\mathcal{C}}^r||_2^2 + \lambda_{\text{mask}}||\mathcal{O}^r - \tilde{\mathcal{O}}^r||_2^2$. We set the loss weights $\lambda_{\text{rgb}}$ and $\lambda_{\text{mask}}$ to 10,000 and 1,000, respectively. Also, we decrease $\lambda_{\text{eik}}$ and $s_{\text{ratio}}$ to 100 and 2, respectively.

### B.3  Evaluation with rich diffusion priors

This part is implemented based on the RichDreamer [34] codebase using the settings of DMTet [40]. The tetrahedral grid resolution is set to 256 and the batch size is set to 4 for two stages. We optimize the geometry for 3,000 steps, with the first 1,000 steps using latent code, followed by an additional 2,000 steps for texture optimization. While RichDreamer reports significantly increased stability at a rendering resolution of 1024, we achieve stable results at a lower resolution of 512. Therefore, we set our rendering resolution to 512.

**CLIP score**   We adopt the evaluation process in RichDreamer [34] to calculate CLIP scores using the CLIP model [35] (vit-g-14). For geometry CLIP scores, we render generated meshes with uniform albedo and produce 16 different views for each object. The average CLIP scores are computed by discarding the highest and lowest scores from the provided text prompts. For texture CLIP scores, textured meshes are rendered. Since RichDreamer has not released the prompt list (113 objects) used for their metrics, we utilize an alternative list named "*prompts_dmtet.txt*" (183 objects) available on their official GitHub repository.

## C   More results

We present an extensive gallery of visual results in Figure 10-13.

## D   Discussions on the potential social impacts

The proposed *TeT-Splatting* method can make it easier for people to enter the animation and related industries. This can simplify production processes, reduce costs, and allow more people to create high-quality 3D assets. However, these improvements could also lead to job losses for professionals who work in traditional roles. To address this, it may be necessary to gradually adjust training programs and align the workforce with future demands.

Additionally, while our method improves the efficiency of 3D generation, it also carries the biases present in the foundational models we use. These models can have built-in biases related to race, gender, and culture, which might appear in the generated content, reinforcing stereotypes. The easier creation of realistic 3D models also raises concerns about copyright infringement and misuse, highlighting the need for strong ethical guidelines and regulatory frameworks.

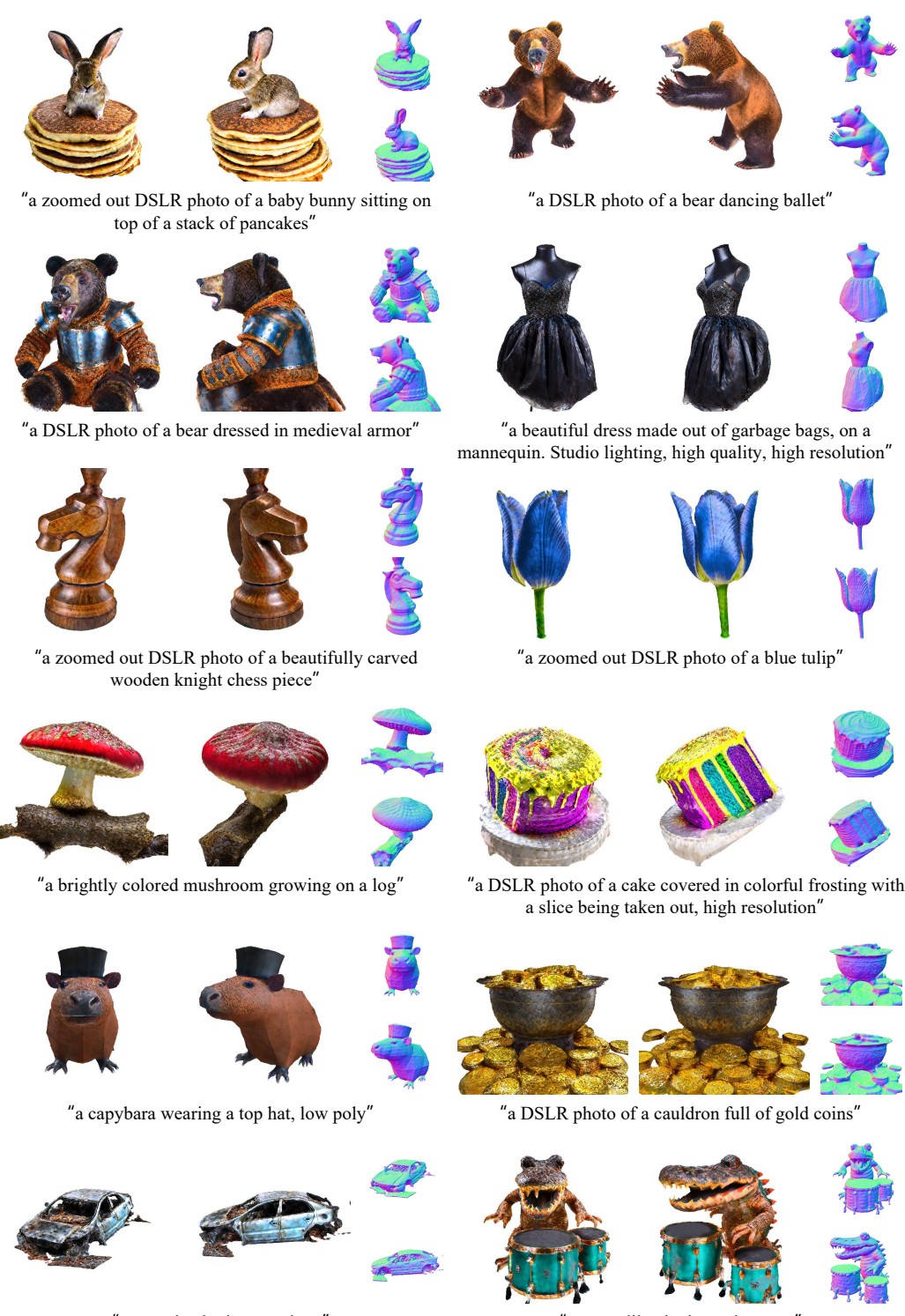

"a zoomed out DSLR photo of a baby bunny sitting on top of a stack of pancakes"

"a DSLR photo of a bear dancing ballet"

"a DSLR photo of a bear dressed in medieval armor"

"a beautiful dress made out of garbage bags, on a mannequin. Studio lighting, high quality, high resolution"

"a zoomed out DSLR photo of a beautifully carved wooden knight chess piece"

"a zoomed out DSLR photo of a blue tulip"

"a brightly colored mushroom growing on a log"

"a DSLR photo of a cake covered in colorful frosting with a slice being taken out, high resolution"

"a capybara wearing a top hat, low poly"

"a DSLR photo of a cauldron full of gold coins"

"a completely destroyed car"

"a crocodile playing a drum set"

Figure 10: More results of *TeT-Splatting* with rich diffusion priors.

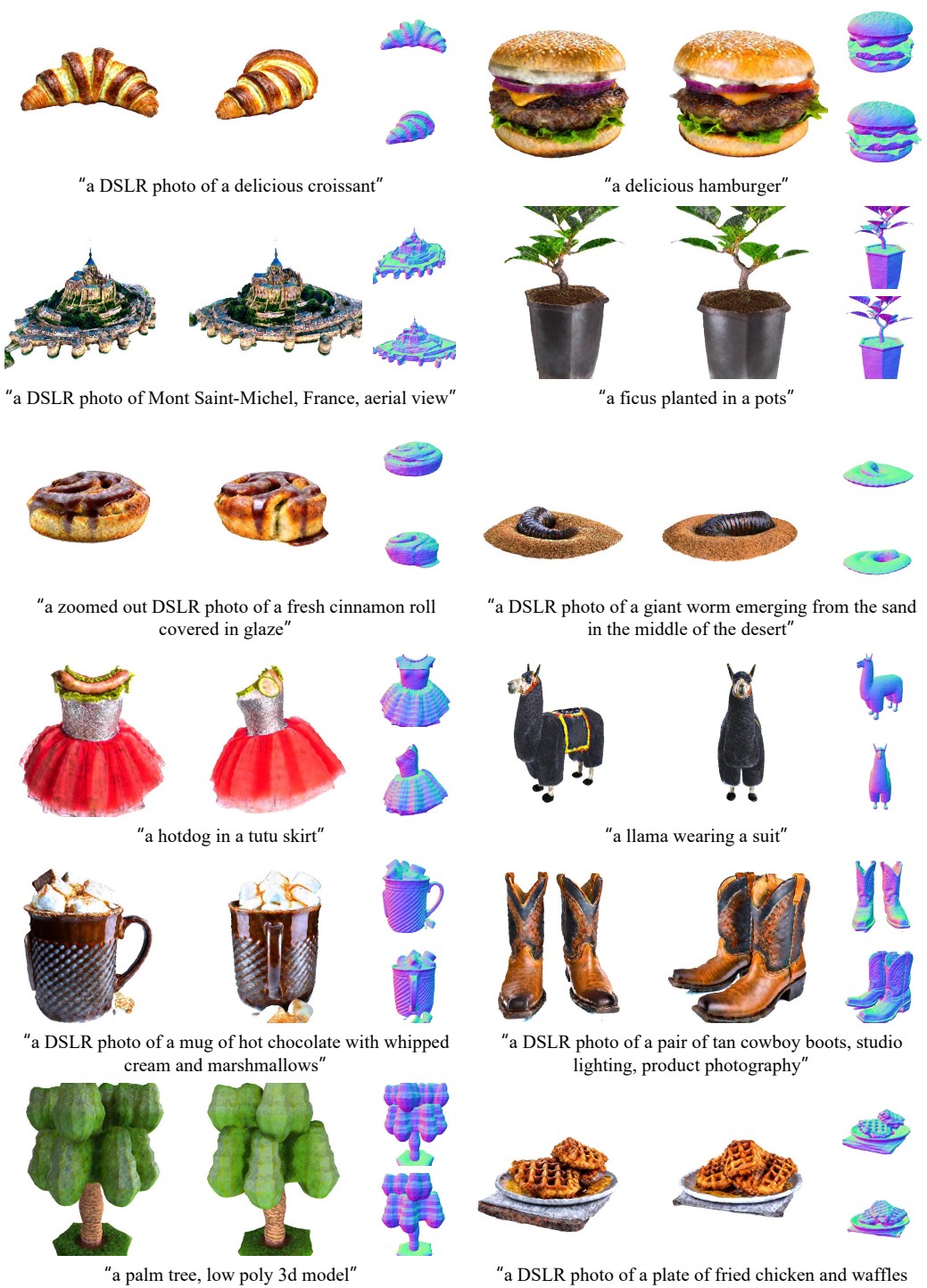

"a DSLR photo of a delicious croissant"

"a delicious hamburger"

"a DSLR photo of Mont Saint-Michel, France, aerial view"

"a ficus planted in a pots"

"a zoomed out DSLR photo of a fresh cinnamon roll covered in glaze"

"a DSLR photo of a giant worm emerging from the sand in the middle of the desert"

"a hotdog in a tutu skirt"

"a llama wearing a suit"

"a DSLR photo of a mug of hot chocolate with whipped cream and marshmallows"

"a DSLR photo of a pair of tan cowboy boots, studio lighting, product photography"

"a palm tree, low poly 3d model"

"a DSLR photo of a plate of fried chicken and waffles with maple syrup on them"

Figure 11: More results of *TeT-Splatting* with rich diffusion priors.

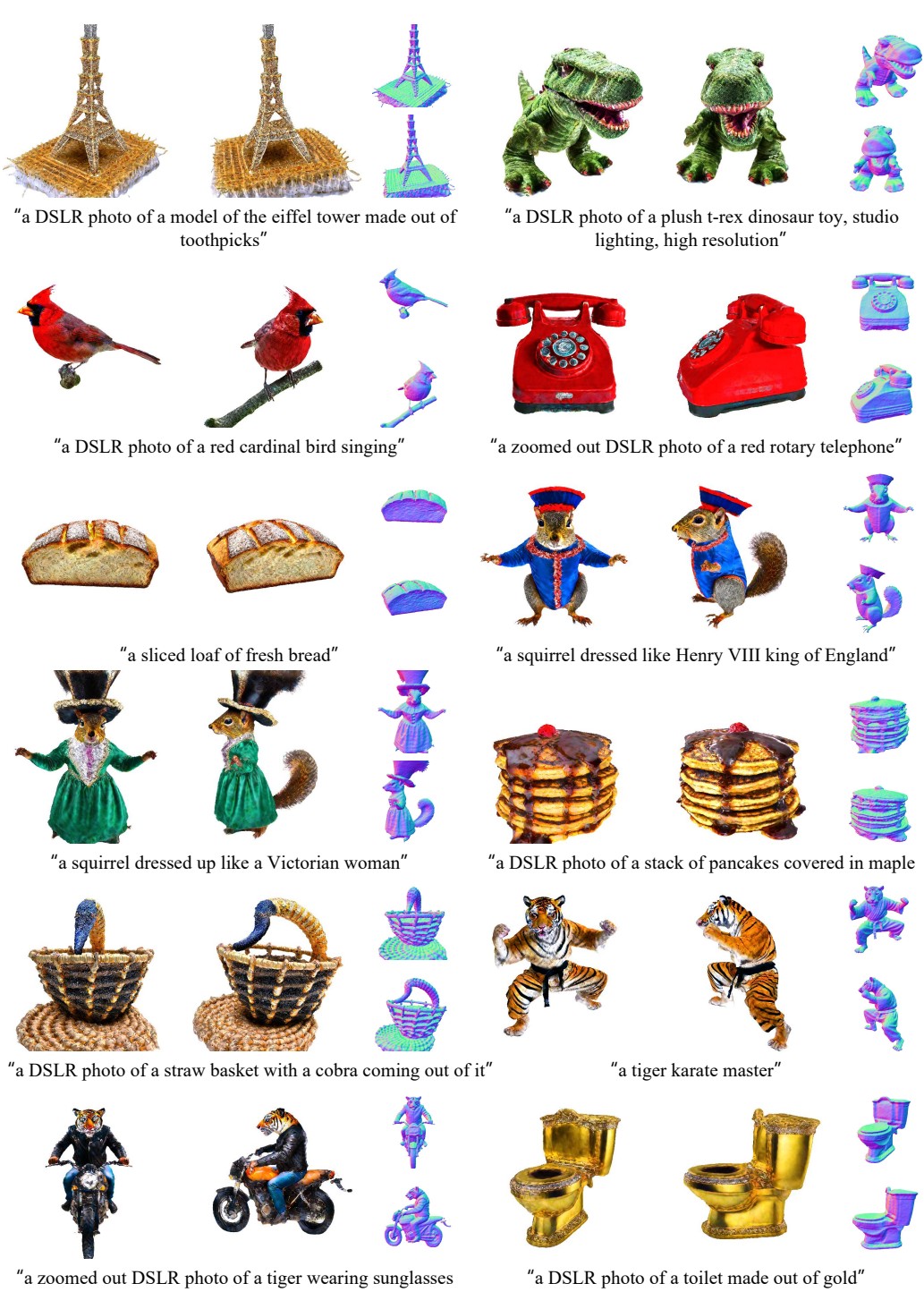

"a DSLR photo of a model of the eiffel tower made out of toothpicks"

"a DSLR photo of a plush t-rex dinosaur toy, studio lighting, high resolution"

"a DSLR photo of a red cardinal bird singing"

"a zoomed out DSLR photo of a red rotary telephone"

"a sliced loaf of fresh bread"

"a squirrel dressed like Henry VIII king of England"

"a squirrel dressed up like a Victorian woman"

"a DSLR photo of a stack of pancakes covered in maple

"a DSLR photo of a straw basket with a cobra coming out of it"

"a tiger karate master"

"a zoomed out DSLR photo of a tiger wearing sunglasses and a leather jacket, riding a motorcycle"

"a DSLR photo of a toilet made out of gold"

Figure 12: More results of *TeT-Splatting* with rich diffusion priors.

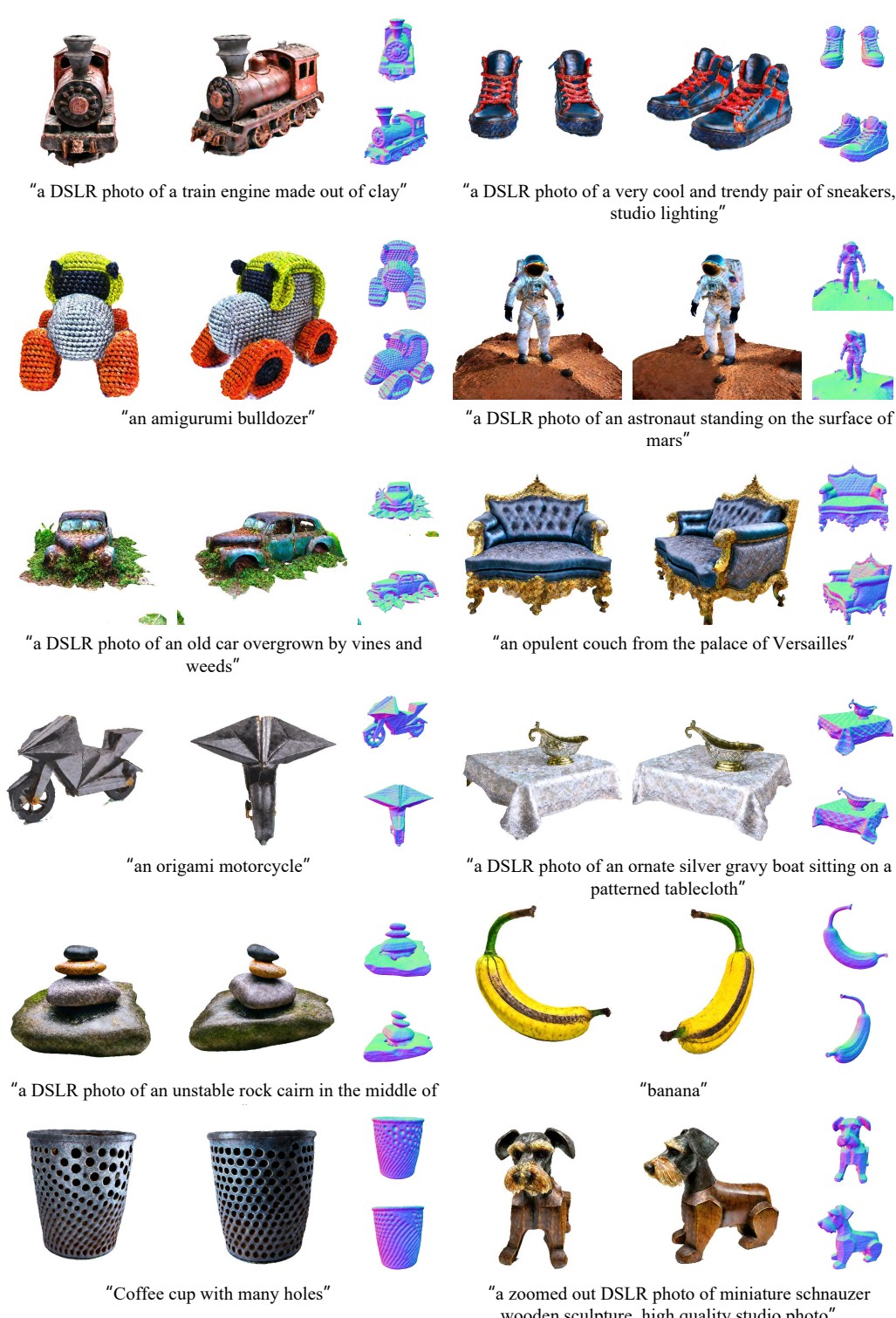

"a DSLR photo of a train engine made out of clay"

"a DSLR photo of a very cool and trendy pair of sneakers, studio lighting"

"an amigurumi bulldozer"

"a DSLR photo of an astronaut standing on the surface of mars"

"a DSLR photo of an old car overgrown by vines and weeds"

"an opulent couch from the palace of Versailles"

"an origami motorcycle"

"a DSLR photo of an ornate silver gravy boat sitting on a patterned tablecloth"

"a DSLR photo of an unstable rock cairn in the middle of"

"banana"

"Coffee cup with many holes"

"a zoomed out DSLR photo of miniature schnauzer wooden sculpture, high quality studio photo"

Figure 13: More results of *TeT-Splatting* with rich diffusion priors.

