# OpenReview forum: "Tetrahedron Splatting for 3D Generation"
_NeurIPS.cc/2024/Conference — NeurIPS 2024 spotlight_

### Official Review · Reviewer_Tjej · 2024-07-12

**Soundness:** 3
**Presentation:** 2
**Contribution:** 2
**Rating:** 6
**Confidence:** 4

**Summary:**

This paper proposes a method for 3D mesh generation from a text. The idea is to improve gradient propagation ability of DMTet framework by splatting multiple tetrahedra instead of rendering only one surface triangle per pixel. Qualitative results on 3D mesh generation show good quality results with competitive processing time.

**Strengths:**

-	The proposed idea is interesting. Splatting multiple tetrahedra around the surface allows to backpropagate gradients further in the model and avoid getting stuck in local minima.
-	Figure 3 is interesting and shows advantage of the proposed framework.
-	I am not so familiar with the datasets and tasks of 3D mesh generation from text but qualitative results look good and competitive with state-of-the-art

**Weaknesses:**

-	Some explanations are missing to fully understand the experiments section.
-	Results are few and maybe cherry picked. Only qualitative cannot prove better performance of the proposed method. Again, more explanations about the generation process is required to understand advantages. For example, in figure 6 the 2D images are different, with different level of details in the 2D images. Then it is expected than the 3D models are also different and level of details also different. Then does it really make sense to compare these output 3D meshes? The difference may come from other parts in the framework than the proposed model.
-	The authors motivate many of text for computational speed. But in the end advantage in computational power is not clear.

**Questions:**

-	How many tetrahedra are used in the experiments?

The proposed idea to splat tetrahedra and improve gradients propagation ability of original DMTet is interesting. Qualitative results show good performance. However, the explanations in the paper of the 3D generation process is lacking. Quantitative evaluation is few and only few qualitative results are maybe not enough to confirm advantage of proposed method.

**Limitations:**

There is no specific section or discussion about the limitations of the proposed method

---

> ### Author Rebuttal · Authors · 2024-08-06
>
> ### **Response to Reviewer Tjej**
>
> Thank you for your valuable feedback and constructive suggestions. We've addressed your concerns in the revised manuscript and our response below.
>
> **Q1: Explanations of pipeline and experiment.**
>
> We apologize for any confusion. Here, we further summarize our pipeline (as stated in Section 4) and experiment settings (as stated in Section 5).
>
> Pipeline: The 3D generation process is decomposed into two stages. In the first stage, we use TeT-Splatting to render normal, depth, and opacity maps from the tetrahedral grid and signed distance field (SDF). We then use SDS loss to guide these maps, resulting in detailed geometry. In the second stage, we fix the geometry and extract the polygonal mesh from the tetrahedral grid and SDF using Marching Tetrahedra. Through rasterization and querying the texture network, we render the albedo and PBR maps, which are again guided by SDS loss.
>
> Experiment: We assess the efficacy of TeT-Splatting using two different diffusion priors: vanilla diffusion priors (leveraging Stable Diffusion as the diffusion model) and rich diffusion priors (leveraging the Normal-Depth diffusion model and Albedo diffusion model proposed in RichDreamer).
>
> For vanilla diffusion priors, we compare our method with the prior art methods that also use only the Stable Diffusion model for a fair comparison. In Figure 4, we present experiments on text-to-3D and image-to-3D tasks. We also evaluate the mesh exportation process in Figure 5.
>
> For rich diffusion priors, Figure 6 compares our method with state-of-the-art methods in text-to-3D, demonstrating that TeT-Splatting can handle high-fidelity 3D generation. In Figure 7, we visualize the normal maps during the early iterations, showing TeT-Splatting's superiority in stable and compact optimization compared to using DMTet (RichDreamer) as the 3D representation. Additionally, we conduct ablation studies in Figure 8 to demonstrate the effectiveness of our design choices in regularizations and hyperparameters. In Table 2, we report the CLIP scores to quantitatively illustrate the geometry and appearance quality of the generated assets.
>
> **Q2: In figure 6 the 2D images are different?**
>
> We apologize for any confusion. As explained in Q1 above, Figure 6 compares those methods on the text-to-3D task, which do not depend on 2D image input. Regarding the concern that "the difference may come from other parts in the framework than the proposed model", we have already conducted dedicated experiments as shown in Figure 4, where all methods use the Stable Diffusion model as the diffusion prior, eliminating any differences in the diffusion prior. When comparing with RichDreamer in a truly fair manner (see Figures 6 and 7), we only substitute the representation in the first stage from DMTet to TeT-Splatting while keeping all the others the same.
>
> **Q3: Computational speed.**
>
> We have extensively tested the computation cost and speed in the paper. In lines 249-254 and Figure 4, we did compare the training and rendering speeds of different methods. Specifically, in cases using vanilla RGB-based diffusion priors, TeT-Splatting demonstrates quick convergence, taking only 15 minutes for Text-to-3D and 5 minutes for Image-to-3D tasks, achieving a rendering speed of 133 FPS. Although TeT-Splatting operates at a lower FPS compared to DMTet (Fantasia3D) and 3DGS (DreamGaussian), this does not significantly hinder the entire generation process, as the primary speed bottleneck lies in the diffusion model. As stated in lines 275-276, employing TeT-Splatting during the geometry optimization stage significantly reduces the generation time from 70 minutes to 40 minutes.
>
> **Q4: Number of tetrahedra.**
>
> For the experiments using vanilla diffusion priors, the resolution of the tetrahedral grid is set to 128 (please see line 486), resulting in 1.5 million tetrahedra initially, which is then reduced down (lines 150-155) to 200k after optimization. For the experiments using rich diffusion priors, the resolution of the tetrahedral grid is set to 256 (line 504), resulting in 12 million tetrahedra initially, and then dropping down to about 1.5 million after optimization. We show that a denser tetrahedral grid can lead to more detailed geometry as expected.
>
> **Q5: Few quantitative evaluation.**
>
> Overall, we follow the previous 3D generation works (e.g., RichDreamer) for quantitative evaluations. This is because in this work we focus on an alternative representation (TeT-Splatting) to be integrated into any optimization-based 3D generation pipelines.  In general, there is no ideal evaluation metrics for 3D generation tasks thus far, and it is an ongoing research problem.
> Due to no ground truth available for 3D generation evaluation, it makes sense that some proxy metrics must be used, such as the CLIP score for evaluating the similarity between the rendered images and the given prompts we are using. Specifically, to assess the geometry quality, we render the extracted meshes with uniform albedo;  For the texture quality, we render the meshes with the generated texture.
>
> **Q6: No specific section about the limitations.**
>
> Due to page limitations, we discuss the limitations of TeT-Splatting in Appendix D (lines 521-527), focusing on three main aspects:
>
> 1) High-Frequency Features: TeT-Splatting struggles with modeling high-frequency features because it uses tetrahedra as rendering primitives, and the resolution of the tetrahedral grid would impose constraints on top.
>
> 2) Rendering Speed: Although TeT-Splatting achieves real-time rendering, its speed is still relatively slower than 3DGS.
>
> 3) Pre-Filter Operation: The current pre-filter operation may not fully exploit the potential of TeT-Splatting in terms of generation quality and speed. A densification process might improve these aspects, which we leave as future work.
>
> We hope our responses address your queries. If you have any further concerns, please let us know.

---

> > ### Comment · Reviewer_Tjej · 2024-08-12
> >
> > I have read the authors' rebuttal and the other reviews. The authors have answered my questions and I am mostly satisfied with the response. This is an interesting paper.

---

> ### Author Response · Authors · 2024-08-12
>
> Dear Reviewer Tjej
>
> We appreciate the reviewer's time for reviewing and thanks again for the valuable comments.
>
> Best wishes
>
> Authors

---

### Official Review · Reviewer_dtJc · 2024-07-12

**Soundness:** 3
**Presentation:** 4
**Contribution:** 3
**Rating:** 6
**Confidence:** 4

**Summary:**

The paper presents a novel method for generating 3D models from the given inputs. The authors, for that purpose, propose the use of tetrahedra instead of 3D Gaussians in the recent proposed Gaussian Splatting framework.

**Strengths:**

The paper proposes the use of TeT Splatting, that are well formulated and demonstrated  throughout the paper. The method allows for precise and fast mesh extraction, that is a well-known limitation of 3d gaussian splatting. The motivation to use TeT Splatting against the well-known DMTet is also well tackled.

**Weaknesses:**

1. For the task of 3D Reconstruction, 3D Consistency is highly important. However, lack of videos of the generated result is a major drawback that highly affects my decision since just two views are not enough to judge the quality of the output and created mesh. This is also a standard in the baseline papers such as DreamFusion that the authors compare with. I am happy to revise my scores otherwise.

2. The paper uses hash grids as described in sec 4.1. Is it really necessary to use that, or how well the method will work without a hash grid or some other encoding such as positional encoding?

**Questions:**

Please see weakness section.

**Limitations:**

The authors provide limitations and potential societal impact in the supplementary pages of their work.

---

> ### Author Rebuttal · Authors · 2024-08-06
>
> ### **Response to Reviewer dtJc**
>
> Thank you for your valuable feedback and constructive suggestions. We've addressed your concerns in the revised manuscript and our response below.
>
> **Q1: Results from multiple viewpoints.**
>
> We apologize for the lack of videos.
> We are not allowed to provide links to external pages.
> So we have provided additional viewpoints of the generated assets shown in the teaser in the attached PDF. Please refer to Figure 1 in the attached PDF. The generated assets are compact, exhibiting descent 3D consistency.
>
> **Q2: Hash grid or some other encoding?**
>
> This component is flexible in terms of choice of encoding algorithm.
> Hash grid is a common choice for modeling the signed distance field (SDF) due to its efficiency, so it is selected in our work. Nevertheless, other methods such as positional encoding with MLPs or assigning a learnable SDF value to each vertex can also be considered. Per this suggestion, we have provided a comparison in Figure 2 of the attached PDF. Hash grids offer more efficient querying, and faster convergence than positional encoding. Compared to assigning learnable SDF values to vertices, hash grids enhance the 3D consistency and smoothness of the model.
>
> We hope our responses address your queries. If you have any further concerns, please let us know.

---

> > ### Comment · Reviewer_dtJc · 2024-08-13
> >
> > I thank the authors for their prompt rebuttal. I shall update my score accordingly.

---

### Official Review · Reviewer_7DJZ · 2024-07-15

**Soundness:** 4
**Presentation:** 3
**Contribution:** 3
**Rating:** 7
**Confidence:** 5

**Summary:**

The paper presents a new differentiable 3D shape representation based on DMTet. It borrows the efficient rasterization-based rendering techniques from Gaussian Splatting to improve the global optimization of DMTet. The new TeT splatting is shown to perform well in the task of text-to-3D.

**Strengths:**

- The paper presents an interesting new differentiable representation for 3D meshes. It is more efficient than DMTet in optimization.

- The representation works well in text-to-3D generation.

**Weaknesses:**

- The comparison in the experiment section is mostly qualitative.  A quantative comparison with DMTet on a 3D mesh reconstruction task would be appriciated.

- The quality of underlying mesh connectivities is unclear to me.

**Questions:**

-  How to ensure the sharpness of texture obtained without extracting mesh during optimization?

- Can the proposed method be used in a feedforward generative model?

**Limitations:**

Yes, the limitation and societal impact discussions look good to me.

---

> ### Author Rebuttal · Authors · 2024-08-06
>
> ### **Response to Reviewer 7DJZ**
>
> Thank you for your valuable feedback and constructive suggestions. We've addressed your concerns in the revised manuscript and our response below.
>
> **Q1: Sharpness of texture.**
>
> As discussed in the limitation section of the Appendix, lines 521-523, TeT-Splatting uses tetrahedra as rendering primitives while modeling the 3D object with a structured tetrahedral grid. The use of a structured tetrahedral grid allows for the direct extraction of polygonal mesh through Marching Tetrahedra. On the other hand, it also limits TeT-Splatting's ability to model high-frequency textures. To address this, we decompose the 3D generation process into two stages in sequence: first, we optimize the rendering normal maps to obtain the geometry, and then we transition to a polygonal mesh through Marching Tetrahedra for further high-frequency texture optimization. We will further clarify and stress.
>
> **Q2: TeT-Splatting in a feedforward generative model.**
>
> Absolutely! Integrating TeT-Splatting into a feedforward generative model as a 3D representation is a compelling concept. For instance, InstantMesh[1] uses a ViT encoder and a tri-plane decoder to generate tri-planes from multi-view images. They point out that volume rendering is memory-intensive, limiting the use of high-resolution images. To address this, they employ a differentiable isosurface extraction method (FlexiCubes[2]) to extract meshes from the tri-plane. However, this method lacks global awareness, as it can only backpropagate to the zero-level set of the signed distance field. This is where our TeT-Splatting can come into play as the 3D representation to overcome these limitations. This is because TeT-Splatting allows for precise mesh extraction, retains global awareness, while enabling real-time rendering. We reckon this integration could significantly enhance the performance and capabilities of the generative model.
>
> **Q3: Quantitative comparison with DMTet on 3D reconstruction.**
>
> As discussed in Q1, TeT-Splatting is not designed to model high-frequency textures as required by 3D reconstruction, but focuses on the geometry stage in 3D generation, where only normal maps are needed as output. Due to the time constraints of the rebuttal period, we plan to investigate this in future work.
>
> **Q4: Quality of underlying mesh connectivities.**
>
> As explained in lines 105-109, we establish a signed distance field (SDF) by interpolating the SDF values within each tetrahedron of a deformable tetrahedral grid, with each vertex's SDF value provided by a continuous hash network. The tetrahedral grid is structured, and its vertices can only deform within a local region, ensuring that tetrahedra do not intersect each other. After the first stage of optimization, we extract the mesh using the Marching Tetrahedra. This process assigns one or two triangles to each tetrahedron that intersects the zero-level set of the signed distance field. Consequently, as long as we maintain a smooth signed distance field, the extracted mesh will not be fragmented. As shown in Figure 7, TeT-Splatting produces a more coherent mesh compared to DMTet, due to the use of alpha-blending and normal consistency loss during optimization which leads to a more compact signed distance field. Additionally, we provide more viewpoints of the generated assets in the teaser in the attached PDF. Please refer to it for further details.
>
> [1] Xu, Jiale, et al. "Instantmesh: Efficient 3d mesh generation from a single image with sparse-view large reconstruction models." arXiv preprint arXiv:2404.07191 (2024).
>
> [2] Shen, Tianchang, et al. "Flexible Isosurface Extraction for Gradient-Based Mesh Optimization." ACM Trans. Graph. 42.4 (2023): 37-1.
>
> We hope our responses address your queries. If you have any further concerns, please let us know.

---

### Official Review · Reviewer_39Vr · 2024-07-15

**Soundness:** 3
**Presentation:** 3
**Contribution:** 3
**Rating:** 7
**Confidence:** 3

**Summary:**

This submission is proposing a new representation for 3D content generation combining the three following benefits:
- easy to optimize,
- real-time rendering,
- allowing the extraction of precise meshes.
Recent advances in novel view synthesis, i.e. NeRF and then 3D Gaussian Splatting (3DGS) are eventually bringing two of these benefits but the lack of quality output meshes still limits a number of applications.

The proposed approach follows the deformable tetrahedral grid approach of DMTet while using it instead for volume rendering and making it suitable for splatting as in 3DGS. Meshes can be easily extracted which facilitates chaining a first geometric optimization stage based on splat rendering with a later mesh-based texturing stage for complete 3D generation of geometry and appearance.

The paper makes the following contributions:
- using a deformable tetrahedral grid rendered via tetrahedra splatting as a new 3D representation,
- adapting 3DGS to work with such new representation with the implementation of a fast differentiable rasterizer for tetrahedra-based splatting,
- leveraging this representation and optimization in a two-stage 3D generation pipeline, first to optimize tetrahedra geometry before optimizing mesh texturing,
- An array of qualitative evaluations against other generative baselines (with some quantitative evaluations), either on image-to-3D or text-to-3D tasks using vanilla RGB diffusion priors or with rich diffusion priors.

**Strengths:**

- This paper proposes to tackle a known shortcoming of 3D generative approaches when leveraging 3D representation from modern novel view synthesis techniques: poor meshed outputs. Identifying a suitable representation that is easy to optimize, enables real-time rendering while allowing mesh extraction is very relevant to the research community, especially to better bridge 3D and generative work.

- The presented approach elegantly solves the issue of DMNet where only part of the representation (near the zero level set) can effectively be updated. This is addressed by turning the optimization into a full volumetric rendering of tetrahedra via an adaptation of 3DGS to different primitives.

- The qualitative evaluations using vanilla RGB-based diffusion priors against the selected comparable techniques: Magic3D, Fantasia3D and DreamGaussian but based on NeRF, DMTet or 3DGS respectively do a rather convincing job to demonstrate the superiority and versatility of the proposed approach (Figure 4) that maintains high mesh quality with real-time rendering speed. Similarly the qualitative results with rich diffusion priors show pretty high surface quality with a reasonable training time,

**Weaknesses:**

- Apart from the CLIP scores of Table 2 (or the training time and rendering speed), the shown results are essentially qualitative. For a stronger comparison against other baselines, it would have been more helpful to try to integrate measures of the quality of the generated surface to demonstrate the benefits of the approach (instead of relying only on normal maps and renders). Also note that the presentation of these qualitative results could be improved by zooming on specific areas with artifacts (or lack or thereof) to better highlight them and guide the reader.

- It seems there is no information on how the tetrahedral grid is initialized (besides its spatial resolution and Figure 3 hinting at a sphere).

- Some suggestions for improvements:
  - l. 107-109: the formulation make it sound like DMTet invented the established marching tetrahedra algorithm - rephrase
  - Figure 2: Geometry Opimization -> Geometry Optimization
  - Figures 4 and 6: it would be helpful to match the orientation for normals across objects (i.e. have consistent coloring for the whole figure)

**Questions:**

- To address the above weakness, do the authors have some measures of surface quality to provide some form of quantitative evaluation against baselines?

---

> ### Author Rebuttal · Authors · 2024-08-06
>
> ### **Response to Reviewer 39Vr**
>
> Thank you for your valuable feedback and constructive suggestions. We've addressed your concerns in the revised manuscript and our response below.
>
> **Q1: Measures of surface quality.**
>
> We thank the reviewer for raising this question. The best results from our method are achieved using the pipeline with rich diffusion priors in the text-to-3D task. Since there is no image input as guidance, we cannot evaluate the quality of the generated assets with metrics such as PSNR or Chamfer distance. Therefore, following the previous work (RichDreamer[1]), we report the geometry CLIP score and appearance CLIP score. As explained in lines 509-515, the geometry CLIP score is calculated by rendering the extracted meshes with a uniform albedo from 16 different views for each object. We believe this metric is sufficient to measure the surface quality of the extracted mesh and its alignment with the input text. Additionally, we have provided more viewpoints of the generated assets shown in the teaser in the attached PDF (Figure 1). Please refer to it for more details.
>
> **Q2: Presentation of these qualitative results**
>
> Thank you for your advice! We will highlight and zoom in on specific areas in the revised manuscript to better view experience and improve the presentation of qualitative results.
>
> **Q3: Initialization of tetrahedral grid.**
>
> As explained in line 198, we initialize the geometry as a sphere. This process involves fitting the signed distance field (SDF) to a sphere, following Fantasia3D[2]. Specifically, we employ the implementation from threestudio[3], where points are sampled in close proximity to the predefined sphere and the ground truth signed distance values are computed to optimize the SDF. Initializing the geometry as a sphere is a common practice in 3D generation methods using Score Distillation Sampling (SDS) loss, as it helps stabilize the generation process.
>
> **Q4: Some suggestions for improvements.**
>
> Thank you for the suggestions! We will revise the paper to improve the writing quality. Also, we will unify the orientation of the normal maps across objects in Figures for consistent visualization.
>
> [1] Qiu, Lingteng, et al. "Richdreamer: A generalizable normal-depth diffusion model for detail richness in text-to-3d." Proceedings of the IEEE/CVF Conference on Computer Vision and Pattern Recognition. 2024.
>
> [2] Chen, Rui, et al. "Fantasia3d: Disentangling geometry and appearance for high-quality text-to-3d content creation." In \textit{ICCV}. 2023.
>
> [3] Liu, Ying-Tian, et al. "Threestudio: A modular framework for diffusion-guided 3d generation." In \textit{ICCV}, 2023.
>
> We hope our responses address your queries. If you have any further concerns, please let us know.

---

> > ### Comment · Reviewer_39Vr · 2024-08-12
> >
> > I have read the authors' rebuttal and the other reviews.
> >
> > I would first like to thank the authors for preparing this rebuttal and for their detailed answers.
> >
> > > Q1: Measures of surface quality.
> > > [...] Since there is no image input as guidance, we cannot evaluate the quality of the generated assets with metrics such as PSNR or Chamfer distance.
> >
> > This was understood, the proposal was actually to explore (and possibly probe via ablations) intrinsic indicators of the surface quality of the produced mesh (e.g. smoothness) given the claims around geometric quality.
> >
> > Anyway, since most of my concerns and questions have been addressed, I will stick to the initial rating I proposed for this submission (Accept).

---

> > > ### Author Response · Authors · 2024-08-13
> > >
> > > Dear Reviewer 39Vr
> > >
> > > We appreciate the reviewer's time for reviewing and thanks again for the valuable comments and the positive score!
> > >
> > > Best wishes
> > >
> > > Authors

---

### Author Rebuttal · Authors · 2024-08-06

We provide two additional figures in the attached PDF:

1) **Figure 1: More viewpoints of the generated assets shown in the teaser.**

2) **Figure 2: Comparison of different encodings used for modeling signed distance field.**

---

### Decision · Program_Chairs · 2024-09-25

**Decision:**

Accept (spotlight)

**Comment:**

All the reviewers liked the submission. Some queries were raised in the initial reviews, but the authors satisfactorily resolved them during the rebuttal phase.

The paper proposes a novel approach that employs a deformable tetrahedral grid for volume rendering and splatting, enabling mesh extraction and optimization for 3D (geometry and appearance) generation. The main contributions, as highlighted in the reviews, are a new 3D representation utilizing tetrahedra splatting, its adaptation with a fast differentiable rasterizer, and its utilization in a two-stage 3D generation pipeline. The method performs effectively in text-to-3D generation tasks, offering high-quality mesh extraction. The technique is clearly explained and extensively evaluated, producing competitive results in 3D mesh generation tasks.